# 3D-Pix: 3D Editing with Single-Shot Multi-View Diffusion and Gaussian Splatting

## Paper ID 22

## Abstract

*The rapid advancements in 3D visual generative AI are driven by improvements in the quality and realism of 2D generative models, alongside recent developments in efficient 3D reconstruction techniques. In this work, we address the problem of 3D editing by developing a consistent multi-view 2D editing model and leveraging 3D reconstruction methods to obtain a 3D representation. Our approach generalizes across various inputs, including renderings of digital 3D assets and turntable videos of real-world objects. Furthermore, this generalization enables our method to be applied as a post-processing step to any existing 3D generative approach, regardless of the underlying geometry representation model.*

*We introduce 3D-Pix, a model that integrates 2D generation with 3D reconstruction to facilitate 3D editing. A key component of our approach is MV Instruct Pix2Pix XL, a modified version of Instruct Pix2Pix [4], designed to generate consistent multi-view images of the same object using the Stable Diffusion XL [37] image generation model. To ensure coherence across multiple views, we employ a novel interpolation mechanism that enables single-inference processing for consistent editing across multiple images. Additionally, we enhance output fidelity by incorporating a super-resolution upscaling step. The geometry of the asset is estimated using a state-of-the-art 3D Gaussian Splatting [22] model. Our proposed 3D-Pix model effectively balances appearance refinement and geometric accuracy, particularly in preserving high-frequency details and achieving high-fidelity results.*

## 1. Introduction

The field of 3D visual generative AI is undergoing significant growth, driven by advances in the quality and realism of 2D generative models, as well as breakthroughs in novel 3D reconstruction techniques, such as 3D Gaussian Splatting [22]. The increasing demand for automation in the

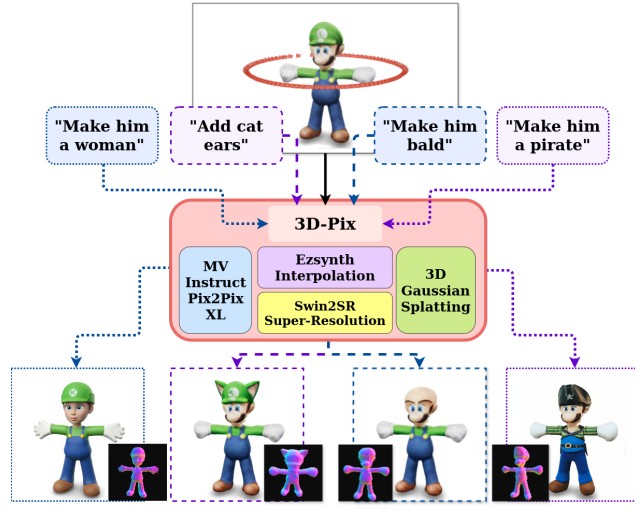

Figure 1. Our 3D-Pix editing high-level process. We expect a turn-around sequence and the edit prompt on input, and our 3D-Pix returns the edited and reconstructed 3D model.

creation of high-quality 3D assets has further accelerated research in this domain.

The 3D generative domain comprises various tasks, commonly classified with respect to input modalities: text-to-3D [8, 32, 38, 43, 43, 47, 58, 70, 71], image-to-3D [12, 27, 29, 49, 50, 61, 72], and 3D-to-3D or editing generations [1, 4, 15, 18, 34, 35]. Although many approaches exist in each direction, many use outdated 2D generative and 3D reconstruction approaches, providing low resolution of the estimated 3D model. Moreover, the 3D-to-3D direction is often treated as re-texturing, aiming to modify only the appearance with no geometry changes.

In this work, we introduce an advanced implicit 3D editing algorithm that operates solely on text prompts, eliminating the need for explicit manual masks or bounding boxes to specify the target region for editing. Our approach presents a novel 3D editing pipeline that integrates sequential components of 3D reconstruction using 3D Gaussian Splatting [22] with a multi-view editing framework. This framework

Figure 2. Our 3D-Pix text-guided edit renderings of the reconstructed 3D models.

leverages the Stable Diffusion XL [37] generative model and the 2D editing capabilities of Instruct Pix2Pix [4], allowing high-fidelity text-guided modifications of 3D assets.

In summary, our contributions are:

- We propose a novel Multi-View Instruct Pix2Pix XL model, a modified version of the original 2D editing model Instruct Pix2Pix [4] to generate consistent multi-view frames of the same object using the SOTA Stable Diffusion XL [37] generative model.
- We propose our 3D editing model 3D-Pix by leveraging our pre-trained Multi-View Instruct Pix2Pix XL model in a single-inference manner due to being coupled with the complex interpolation logic, together with the 3D Gaussian Splatting [22] geometry reconstruction.
- We demonstrate the generalization of 3D-Pix to various real-life and digital inputs and its application as a post hoc stage for an arbitrary 3D generative model of any geometry representation technique.

## 2. Related Works

### 2.1. Diffusion Models

Although diffusion generative models were first introduced by Sohl-Dickstein *et al*. [45], they gained widespread attention following the DDPM work [20], which significantly improved diffusion processes and sampling strategies. DDIM [46] further enhanced efficiency by introducing a non-Markovian denoising process, enabling faster sampling with superior quality.

Although diffusion models eventually surpassed GANs in quality [14], they remained computationally expensive due to their high-dimensional pixel-space operations. The Latent Diffusion Model (LDM) [39] addressed this by incorporating a VAE, reducing dimensionality and improving efficiency.

Stable Diffusion (SD) [39] became the most widely adopted LDM implementation. Despite numerous advancements [21, 33, 60, 69, 77], the most recognized diffusion model today is Stable Diffusion XL [37], which achieves state-of-the-art realism and quality.

### 2.2. 3D Representation Models

When estimating a 3D model from a set of images, geometry can be represented using explicit or implicit techniques. Explicit representations are widely used due to their simplicity and efficiency, with common approaches including 3D voxels [9, 41, 67], meshes [16, 17, 36, 56], and point clouds [26, 48, 63, 73].

Implicit representations are also widely adopted, with many works leveraging signed distance functions (SDF) [7, 42, 44, 64] and occupancy fields [28, 74, 75]. However, NeRF [30] and its extensions [2, 6, 13, 29, 54, 55, 68] dominate much of the research in implicit 3D modeling.

Despite extensive efforts to accelerate NeRF, even the latest optimizations remain computationally demanding. A significantly faster alternative is 3D Gaussian Splatting (3D GS) [22], which not only surpasses NeRF in speed but also achieves superior visual quality while enabling real-time rendering. Due to its efficiency, 3D GS is increasingly being adopted in 3D generation tasks [8, 23, 24, 31, 62, 65].

### 2.3. Text-to-3D

One of the most influential works in text-to-3D generation using diffusion models is DreamFusion [38], which introduced the Score Distillation Sampling (SDS) loss. This novel loss function operates in parameter space, using a frozen diffusion model as a critic to guide NeRF optimization. SDS remains a fundamental component in many modern 3D generation methods [43, 47, 58, 70, 71].

A common strategy is training a multi-view diffusion, as introduced in MVDream [43], to generate consistent multi-views [51]. GSGEN [8] is among the first works to integrate 2D diffusion-based generation with 3D Gaussian Splatting, coupled with pre-trained text-to-point-cloud diffusion Point-E [32] and effective guidance of the geometry estimation of 3D Gaussians using 3D SDS loss.

## 2.4. Image-to-3D & Re-texturing

The re-texturing problem differs from full 3D reconstruction as it often requires no geometry changes and can be addressed by generating PBR materials [25, 66]. However, these methods may not apply to implicit geometry, necessitating further optimization of the 3D model. Many image-to-3D models can also be adapted for re-texturing tasks [29, 72].

Latent-NeRF [29] was among the first diffusion-based methods to generate 3D objects using both text and image inputs, introducing a model for retexturing based on pattern images. IP-Dreamer [72] expanded this approach, being the first to implement Image Prompt (IP) control in Stable Diffusion with modifications to the SDS loss. Several other methods can generate 3D models from a single image [12, 27, 49, 61], with DreamGaussian [50] being the most relevant, as it utilizes 3D Gaussian Splatting for reconstruction.

## 2.5. 3D Editing

The editing task gained significant popularity following the Prompt-to-Prompt work [19], which eliminated the need for manual mask selection and introduced edits controlled solely by text prompts through attention mechanisms. This innovation has spurred further research in the domain [1, 4, 18, 35].

Recent approaches that leverage 3D Gaussian Splatting for geometry optimization include GSEdit [34], GaussianEditor [15], View-consistent Editing (VcEdit) [57] and GaussCtrl [59] models. GSEdit [34] iteratively guides the reconstruction process using Score Distillation Sampling (SDS) loss with Instruct Pix2Pix as the diffusion model, allowing refined edits based on user prompts. GaussianEditor [15] separates editing tasks into object removal and incorporation through semantic tracing, followed by Hierarchical Gaussian Splatting (HGS). VcEdit [57] introduces 3DGS coupled with Cross-attention and Editing Consistency modules to improve multi-view consistency. GaussCtrl [59] employs depth guidance with ControlNet [76] to enhance geometric consistency and the attention-based latent code alignment module to improve texture consistency.

## 3. Preliminary

### 3.1. Instruct Pix2Pix

Instruct Pix2Pix [4] is a state-of-the-art (SOTA) 2D diffusion model designed for text-guided image editing. The authors introduced a novel training methodology based on a synthetically generated dataset, leveraging Prompt-to-Prompt [19] and the GPT-3 language model [5] to create text-image editing pairs.

The dataset generation process consists of three key stages. First, an "edited prompt" is generated by conditioning GPT-3 on the original image description and the given editing instruction, producing a modified description that aligns with the intended transformation. In the second stage, Stable Diffusion [39] and Prompt-to-Prompt [19] are employed to generate both the original and edited images corresponding to the prompts.

The model is trained by minimizing the latent diffusion objective function, conditioned on text input $c_T$ and the image $c_I$:

$$L = \mathbb{E}_{\mathcal{E}(x),\mathcal{E}(c_I),c_T,\epsilon,t}\left[||\epsilon - \epsilon_\theta(z_t, t, \mathcal{E}(c_I), c_T)||_2^2\right], \quad (1)$$

where $z_t$ is the noisy latent variable after diffusing for $t$ steps the input image $x$ in a latent space with the encoder $z = \mathcal{E}(x)$.

## 4. Approach

Generation consistency for many images was proven to be a challenging task ([43]). Unlike GSEdit [34], which used Instruct Pix2Pix edits iteratively for geometry optimization with SDS loss, we edit the frames in only a single inference with our modified Instruct Pix2Pix model, making 2D edits and 3D reconstruction as independent processes.

The main idea of our approach is to process the edits described in the input prompt with a single inference of the generative model to achieve consistency. For that, we divide all of the input frames into specifically four orthogonal key frames and the rest of the intermediate (inter) frames. The key images capture the most information about the object from different angles, and the edits with the diffusion model are applied only to those with our proposed Multi-View variation of the Instruct Pix2Pix model [4] (see Sec. 4.2). The rest of the frames are edited by interpolating the edited key frames into the poses of the original inter frames (see Sec. 4.3) to achieve edits consistent with the key ones. The full pipeline is shown in Fig. 4. This idea is highly inspired by the 3D-GSR [3] work, which leverages consistent 3D Super-Resolution by leveraging 2D Super-Resolution and 3D GS models.

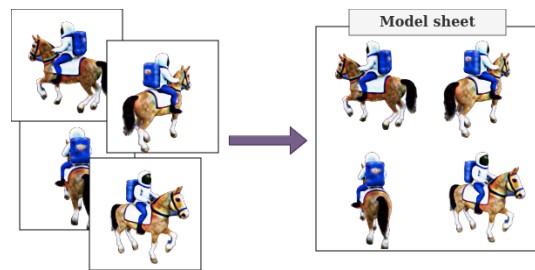

Figure 3. A model sheet is a single image composed as a grid of the orthogonal frames of the same object.

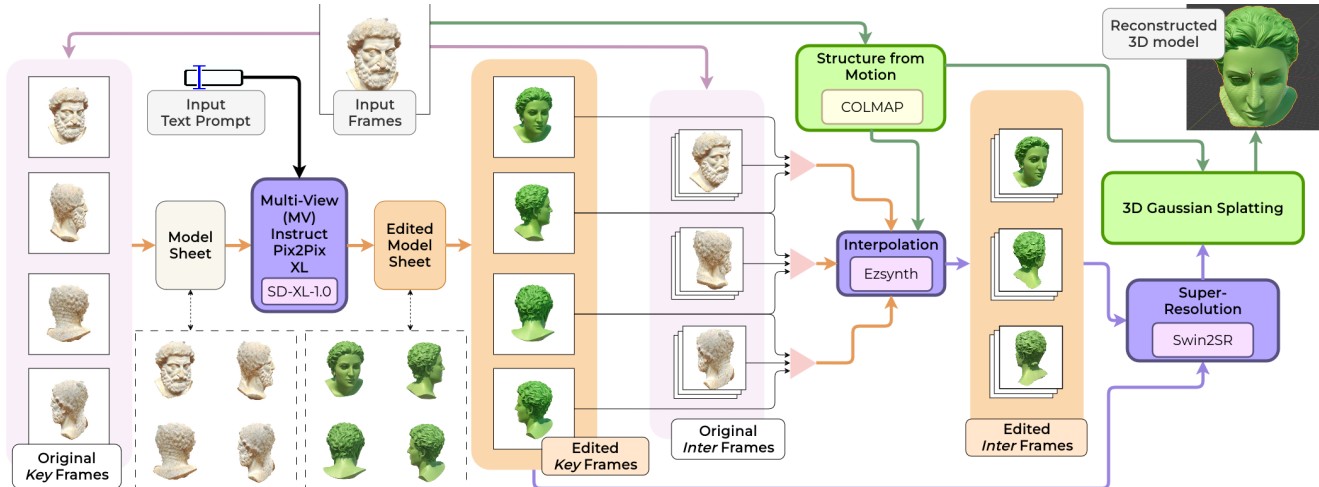

Figure 4. The complete architecture of 3D-Pix. Our proposed MV Instruct Pix2Pix XL is used to edit several key frames from the input sequence given the prompt guidance, and the complex interpolation algorithm is used to achieve the edits of all intermediate frames consistent with the already changed images. Once all of the frames are edited, the images are upscaled with a Super-Resolution model and passed to the 3D GS reconstruction together with the SfM point cloud created from the original sequence.

## 4.1. Model sheet

The idea is first introduced in MVDream [43] work, where the authors trained a multi-view diffusion model to generate four orthogonal views as a grid. Similar to their approach, we also compose a grid of orthogonal frames as in Fig. 3, which we call a model sheet, as the SD produces consistent generations when done in a single inference.

## 4.2. Multi-View Instruct Pix2Pix XL

Instruct Pix2Pix [4] is a high-fidelity, text-guided image editing model built upon the original Stable Diffusion v1 framework [39], capable of generating images at a resolution of $512 \times 512$.

To enhance both the quality and resolution of 2D-generated outputs, we adapt the training methodology of Instruct Pix2Pix to a more advanced model—Stable Diffusion XL (SDXL) [37], which is approximately three times larger in scale. Specifically, we follow the modified training instructions provided in the official implementation (https://github.com/huggingface/diffusers/blob/main/examples/instruct_pix2pix/train_instruct_pix2pix_sdxl.py) and fine-tune the SDXL-base-1.0 checkpoint as our foundation.

Ensuring consistency in prompt-guided edits across multiple frames of a given sequence presents a key challenge. To address this, we employ a model sheet approach, wherein multiple key frames are aggregated into a single composite image (sheet). Edits are then applied in a single inference pass, preserving temporal and structural coherence across frames. To facilitate this, we modify the data

generation process for Instruct Pix2Pix while maintaining the original training pipeline, introducing Multi-View Instruct Pix2Pix XL (MV Instruct Pix2Pix XL) for multi-view editing.

For our dataset construction, we utilize Objaverse 1.0 [11], a large-scale collection of over 800K 3D models. To optimize computational resources—given the high cost of SDXL fine-tuning—we filter the dataset to include only high-definition (HD) models, reducing it to approximately 50K assets. From each selected 3D model, we generate random renderings from four orthogonal viewpoints, with an initial camera position randomly assigned. These renderings are then composed into model sheets, with each 3D asset yielding 10 different sheets from diverse perspectives, resulting in a total of 40 individual renders per object. Each model sheet is further paired with a unique editing prompt, ultimately producing 500K training samples.

Following the methodology outlined in the original Instruct Pix2Pix paper, we leverage the GPT-3 model [5] to generate triplets of text prompts. Each triplet consists of (1) an image caption, (2) an edit instruction, and (3) a caption describing the modified image. These structured prompts are then used in conjunction with Stable Diffusion and Prompt-to-Prompt [19] to generate the corresponding image edits.

More specifically, for the input model sheet $m$ composed of 4 key frames, the forward pass of the SD-XL model adds noise to the encoded latent variable $z = \mathcal{E}(x)$ and produces the noisy variable $z_t$. We learn a network $\epsilon_\theta$ to predict the noise added to the diffused latents $z_t$, conditioned with the edited by Prompt-to-Prompt [19] model sheet $c_M$ and the text prompt $c_T$, by optimizing the conditioned latent diffu-

sion loss function:

$$L = \mathbb{E}_{\mathcal{E}(m),\mathcal{E}(c_M),c_T,\epsilon,t}\left[||\epsilon - \epsilon_\theta(z_t, t, \mathcal{E}(c_M), c_T)||_2^2\right]. \quad (2)$$

While the generation process ensures consistency within a single model sheet, it does not guarantee coherence across multiple sheets when edited in separate inference passes. If two model sheets were edited independently, the resulting modifications could diverge, leading to inconsistencies across frames. To maintain uniformity across all original input frames, we propose interpolating the edited outputs of a single model sheet rather than generating new sheets for each edit. This approach prevents discrepancies between successive generations and ensures a more temporally and structurally consistent editing process.

### 4.3. Interpolation

Since MV Instruct Pix2Pix XL generates edits for only a subset of frames from the model sheet, it is necessary to propagate these modifications to the remaining input images. To achieve this, we interpolate the content of the edited model sheet frames into the corresponding unmodified poses, effectively transferring the new edits to the entire sequence. The interpolation process between two key frames is illustrated in Fig. 6.

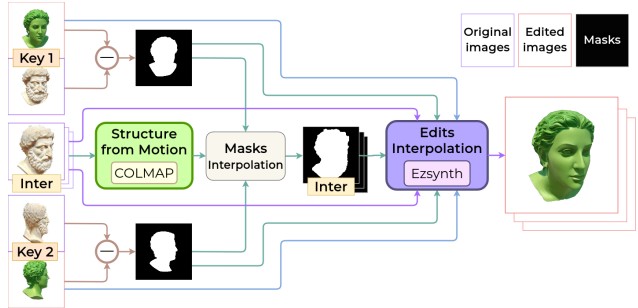

Figure 6. Interpolation process of a single intermediate frame batch given two key frames. Firstly, we classically interpolate only the binary mask of the area to apply edits with some margin, and then the Ezsynth [53] model is used to generate the edited images, given the updated content from the key frames.

For this interpolation, we employ the Ezsynth video stylization model [53], which is based on the Recurrent All-Pairs Field Transforms (RAFT) model for optical flow [52]. In our pipeline, we treat Ezsynth as a black-box component, fine-tuning its hyperparameters to reduce reliance on edge detection, thereby allowing for more significant geometric transformations in the inputs.

To ensure high-fidelity interpolation, we constrain the model to modify only the regions that were edited by MV Instruct Pix2Pix XL. For each input frame, we generate a binary mask highlighting the areas requiring interpolation. This is accomplished by first identifying the modified regions in the model sheet frames by computing the difference between the original and edited images and thresholding the result to create four binary masks. To refine these masks, we apply morphological opening followed by closing operations.

Additionally, since our pipeline incorporates a Structure from Motion (SfM) step for geometry estimation (see Sec. 4.5), we leverage the matched keypoints and estimated homography between sequential frames. This allows us to warp and transform the masks derived from edited model sheets, generating corresponding masks for the remaining frames. These refined masks are then used as input to the interpolation model, enabling it to synthesize realistic and spatially consistent modifications. This approach produces more coherent and visually accurate interpolated frames than directly transforming the edited regions.

More specifically, having the key frames $k_1' = \epsilon_\theta(k_1)$ and $k_2' = \epsilon_\theta(k_2)$ edited with the MV Instruct Pix2Pix XL model $\epsilon_\theta$, we can describe the interpolating process as an inference of the learned model $\epsilon_\psi$ used to edit the original intermediate frame $i$ into edited frame $i'$:

$$i' = \epsilon_\psi(i, m_i, k_1', k_1 - k_1', k_2', k_2 - k_2'), \quad (3)$$

where the $m_i$ mask of the interpolated image $i$ is obtained via computing the homography matrix $H$ between key and inter frames: $m_i = H^{-1}i$.

The key contribution of this component is that it ensures consistency between intermediate frames and the already edited key frames generated by MV Instruct Pix2Pix XL. As a result, the final output maintains a unified and seamless sequence, preserving both geometric and visual coherence throughout the edited frames.

*"Make astronaut ride a huge cat"*

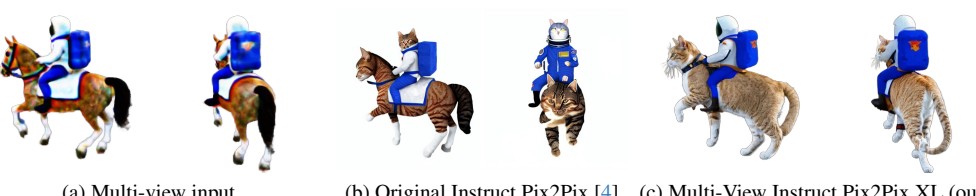

(a) Multi-view input      (b) Original Instruct Pix2Pix [4]    (c) Multi-View Instruct Pix2Pix XL (our)

Figure 5. 2D editing comparison on a simple model sheet of 2 images (a) with original Instruct Pix2Pix (b) and our MV model (c).

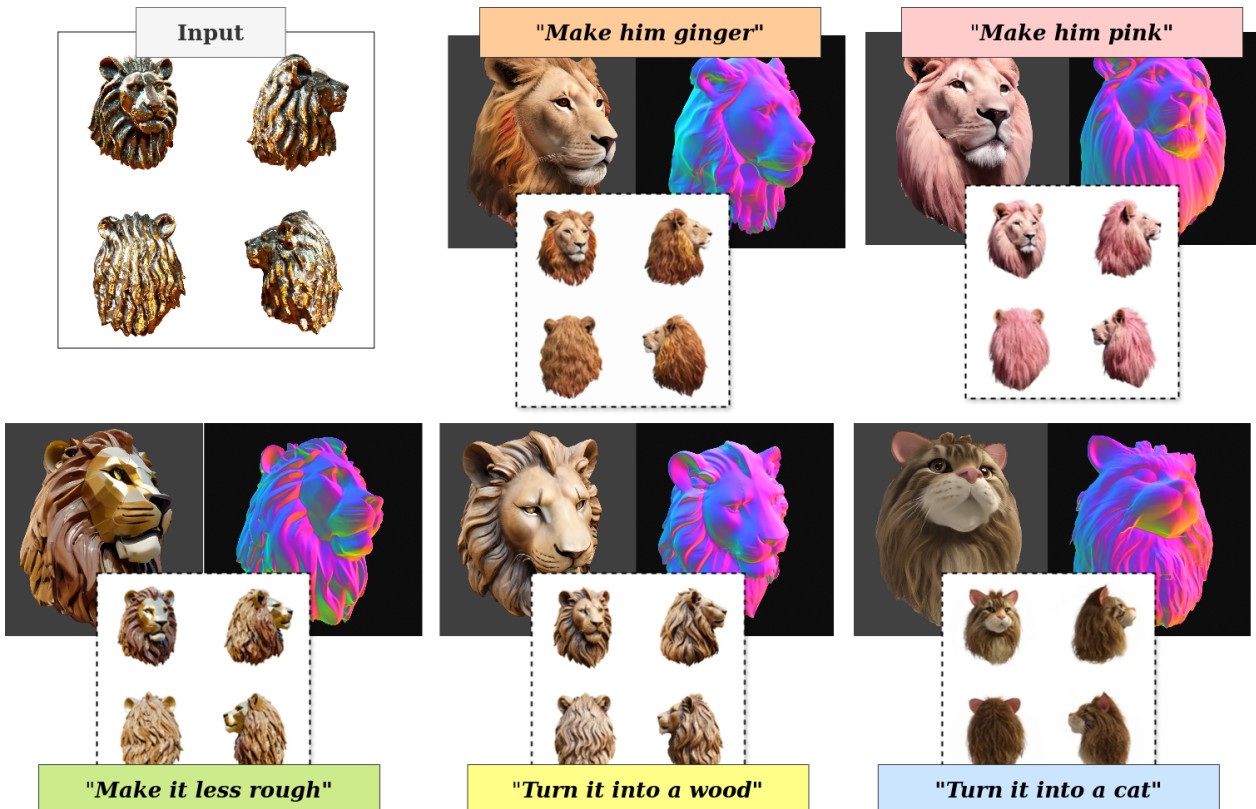

Figure 7. Results of edits with 3D-Pix. The input sequence of a lion is being edited to new 3D models with respect to the input prompts. Our model is capable of producing complex geometry changes (bottom row with structure changes) and appearance changes (top row with color changes).

### 4.4. Super-Resolution

To further enhance the visual fidelity and detail of the images, we integrate an additional Super-Resolution upscaling model, processing frames in batches of four. For this purpose, we employ the transformer-based Swin2SR model [10].

### 4.5. Structure from Motion

Since 3D Gaussian Splatting (3D GS) requires an initial sparse point cloud, and our interpolation pipeline (see Sec. 4.3) relies on estimated homographies between frames, we utilize a classical Structure from Motion (SfM) approach from the COLMAP library [40].

The SfM reconstruction process is applied exclusively to the original, non-edited images. Running SfM on the edited images often leads to failures due to minor inconsistencies introduced during the editing process, as the traditional COLMAP pipeline lacks robustness against such artifacts. However, performing SfM on the original images consistently succeeds, providing a reliable foundation for subsequent 3D GS optimization with an initial coarse set of features.

### 4.6. 3D Gaussian Splatting

For the final 3D reconstruction, we employ the original implementation of 3D Gaussian Splatting [22] as a black-box model. The optimization process refines the initial Gaussians obtained from the sparse point cloud to align with the updated, edited appearance. Additionally, we disable the model's ability to represent view-dependent colors via spherical harmonics (SH) coefficient optimization to maintain consistency in color representation.

## 5. Experiments

### 5.1. Instruct Pix2Pix vs Ours

We evaluate the performance of our MV Instruct Pix2Pix XL model against the original Instruct Pix2Pix in a 2D editing task using a model sheet. A visual comparison is presented in Fig. 5, demonstrating that our proposed approach achieves greater consistency across different frames. In contrast, the original Instruct Pix2Pix struggles to maintain high-quality and consistent modifications, even in a simple case of two sequential images.

To quantitatively assess the differences, we compute av-

Figure 8. Comparison oft our 3D-Pix and other open-source models: GaussianEditor [15] and GaussCtrl [59]. While our model performs more robust color changes, other models perform scene-level edits, while we focus only on object-level reconstruction.

erage CLIP scores on our dataset and report the results in Tab. 1. The evaluation considers two key metrics: (1) Similarity to the input image, ensuring that the edited image remains structurally consistent with the original, and (2) Text-image alignment, measuring the adherence of the edited image to the input textual prompt.

For Instruct Pix2Pix, we test two scenarios: processing a model sheet (similar to our method) and editing individual frames sequentially (which aligns more closely with the original model's training distribution). Our method outperforms both cases in terms of text-image alignment, indicating that it generates more robust and realistic edits. However, our approach yields lower similarity scores to the input images, suggesting that it introduces more substantial modifications compared to the original Instruct Pix2Pix.

## 5.2. Results

We present a diverse set of edited 3D assets generated with 3D-Pix in Figs. 1, 2, 5, 7 and 9. Our approach demonstrates the ability to produce high-quality 3D edits that accurately reflect the desired modifications specified by the text prompt.

We compare our 3D-Pix model with other works in 3D editing with open source code: GaussianEditor [15] and GaussCtrl [59] in the Fig. 8. While our solution provides more robust edits, we perform strictly object-level edits, as opposed to other models capable of generalizable scene-level reconstructions.

| Instruct Pix2Pix model | $CLIP_1 \uparrow$ | $CLIP_2 \uparrow$ |
|---|---|---|
| Our (model sheet) | 0.86 | **0.14** |
| Original (model sheet) | 0.79 | 0.09 |
| Original (frame-wise) | **0.91** | 0.13 |

$CLIP_1$ : Input image similarity
$CLIP_2$ : Edited text - edited image similarity

Table 1. Comparison of CLIP text and image alighnment scores on edited model sheets with our MV Instruct Pix2Pix XL and the original model [4].

Our method is effective for both geometric transformations (e.g., converting an object into a different form) and appearance modifications (e.g., style or color adjustments). The generated assets exhibit high fidelity and successfully capture complex geometric structures while maintaining consistency across multiple views.

## 5.3. Real-Life Inputs

To further evaluate our model's performance, we apply 3D-Pix to turntable-style images captured from real-world objects, as shown in Fig. 9. However, the results indicate a performance degradation compared to digital input data.

Upon analysis of the intermediate outputs, we identify the primary cause as suboptimal key frame selection, resulting from instability in video recordings, abrupt camera movements, or sudden shifts in the object's position. To enhance robustness against such real-world artifacts, we propose integrating a more advanced key frame selection algorithm and applying frame deblurring techniques in future work to improve editing quality.

## 5.4. Failed Cases

Despite its improvements, our approach inherits certain failure cases from the original Instruct Pix2Pix [4], as illustrated in Fig. 10. In some instances, the model fails to isolate the specified object components accurately. For example, in the ice cream scenario, the model erroneously modifies the green jam instead of the intended waffle. Similarly, in the goblet case, it struggles to preserve the fine structure of the input skeleton, resulting in inconsistencies in bone articulation and subsequent reconstruction artifacts.

Furthermore, our 2D editing and interpolation pipeline occasionally misidentifies elements of an object. For example, the top of a glass is misclassified, leading to a non-transparent blue coloration. Additionally, the model fails to correctly interpret the internal composition of a goblet, mistakenly placing a cherry at its center rather than modifying its interior as intended.

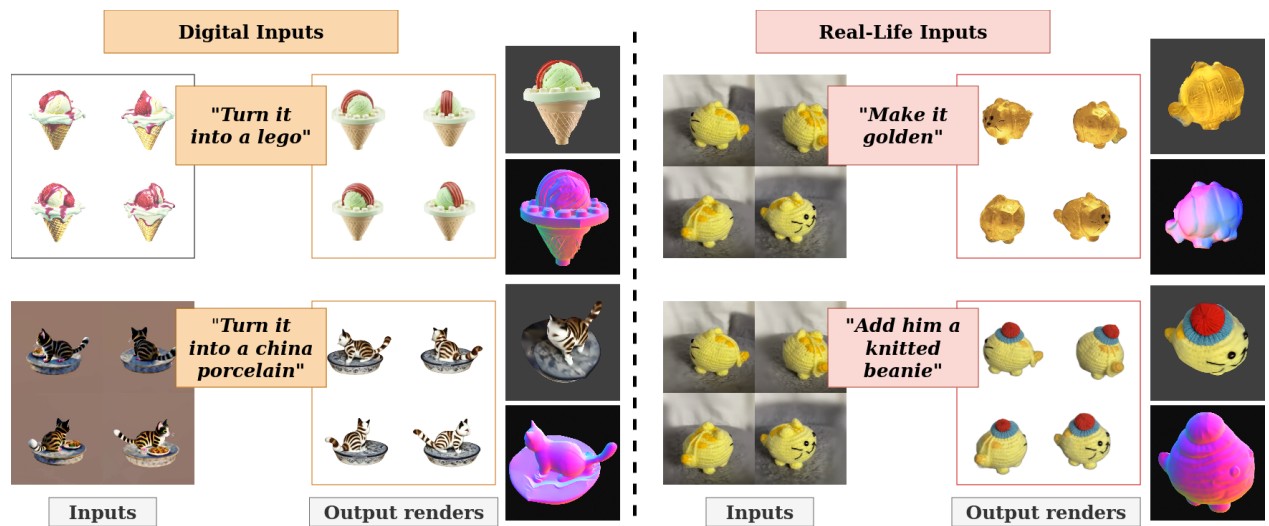

Figure 9. Our 3D-Pix generalizes to both digital (left column) and real-life (right column) inputs.

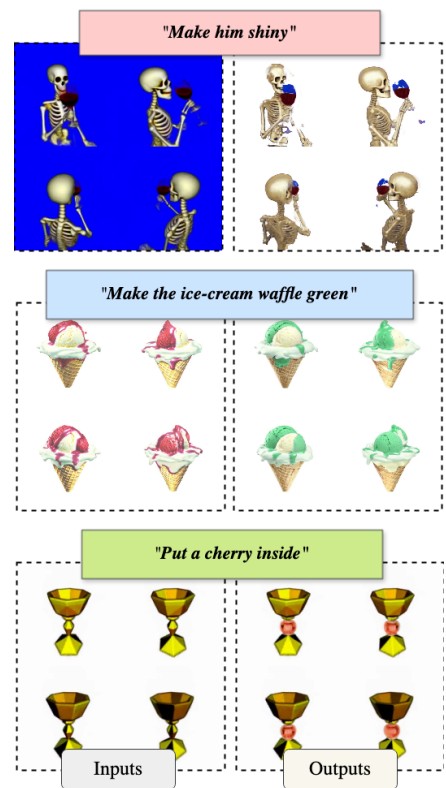

Figure 10. Failed cases. 3D-Pix sometimes struggles with isolating a correct area of the edits (middle and bottom row) or struggles in the thin edges and semi-transparent objects as in the top row.

## 6. Conclusions

In this work, we introduce MV Instruct Pix2Pix XL, an enhanced version of the Instruct Pix2Pix [4] diffusion model, adapted for multi-view image editing. Our approach leverages the state-of-the-art (SOTA) SD-XL [37] generative model, extending its capabilities to ensure consistent modifications across multiple viewpoints.

We further demonstrate how MV Instruct Pix2Pix XL enables 3D editing within our 3D-Pix pipeline, a novel framework designed for efficient multi-view object modification. Unlike conventional methods that require multiple inference passes, our approach performs a single inference step, followed by a complex interpolation process to propagate edits across all views while maintaining consistency.

To enhance the visual fidelity of the outputs, we incorporate the Swin2SR [10] super-resolution model for fine-grained image refinement. Additionally, for geometry estimation, we integrate a Structure from Motion (SfM) pipeline, followed by 3D Gaussian Splatting [22], enabling high-quality 3D reconstruction of the edited asset.

Our method is generalizable to both digital and real-world inputs and can serve as a post-hoc refinement stage for any existing 3D generative model, regardless of its underlying geometry representation.

Beyond the scope of 3D editing, our work raises broader questions about adapting 2D generative models to multi-view settings in a structured and scalable manner. We introduce a novel frame interpolation technique, ensuring that modifications applied to a subset of frames are seamlessly propagated, preserving object consistency across multiple perspectives.

This research paves the way for future explorations in multi-view generative modeling, extending beyond editing to tasks such as multi-view synthesis, reconstruction, and consistency-aware generation.

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
