# OpenReview forum: "3D-Pix: 3D Editing with Single-Shot Multi-View Diffusion and Gaussian Splatting"
_thecvf.com/CVPR/2025/Workshop/CVEU — CVPR 2025_

### Official Review · Reviewer_3gB3 · 2025-03-13

**Rating:** 4
**Confidence:** 4

**Review:**

Summary
This paper presents a 3D editing system with single-shot multi-view diffusion and Gaussian splatting. To this end, MV Instruct Pix2Pix XL is introduced, a modified version of the original Instruct Pix2Pix, designed to generate consistent multi-view images of the same object using Stable Diffusion XL. For the multi-view consistency of generated images, an interpolation mechanism that enables single-inference processing is proposed.

Strengths
3D editing in the 2D space is quite practical and easy to use. The overall system makes sense and the final results have reasonable quality.

Weaknesses
Overall pipeline is highly motivated by previous works, such as MVDream and Instruct Pix2Pix, leading to lack of novelty. Also, there are no video results, which is quite useful to figure out artifacts from these kinds of generated results. Finally, as the system requires quite many components, such as MV Instruct Pix2Pix, COLMAP, super-resolution, and 3DGS, the inference time should be very long.

---

### Official Review · Reviewer_YN1E · 2025-03-18
**Review for Submission22**

**Rating:** 4
**Confidence:** 4

**Review:**

Strengths:

1. The problem is meaningful and important.

2. The proposed system is reasonably effective, though not always perfect in real world scenarios.

3. The provided qualitative and quantitative results are informative


Limitations:

1. Model sheet selection: The paper’s description on how the views for the model sheet is selected is brief. In synthetic data these views may suffice, but in real world data these might not be enough to cover all the 3D information in the model and serve as the basis for 3D consistent interpolation.

2. Reliance on Ezsynth: The approach heavily relies on Ezsynth to maintain 3D consistency. More analysis on how robust Ezsynth is in these scenarios should be provided.

Overall I believe this paper focuses on a meaningful topic, proposes an adequate solution, and demonstrate impressive results. I recommend for weak acceptance.

---

### Official Review · Reviewer_mFPJ · 2025-03-24

**Rating:** 4
**Confidence:** 4

**Review:**

**Strengths**

The qualitative performance is commendable in both appearance and geometry editing, closely aligning with textual input while maintaining high fidelity and coherence. The discussion of failure cases is also adequately addressed.

**Weakness**

1. The use of the CLIP score for evaluation is insufficient due to its inherently vague similarity measure. Incorporating a user study or a GPT-based evaluation could provide a more comprehensive assessment regarding text alignment and quality.

2. The algorithm appears somewhat overly systematic, and Figure 4 is challenging to interpret.

**Question**

Section 4.4, which covers the super-resolution (SR) process, lacks sufficient detail and clarity. Applying independent SR to each frame could introduce 3D inconsistencies during Gaussian Splatting optimization in the final step. Could the authors clarify how this potential issue is addressed?

**Justification**

Overall, despite the limitations in the quantitative evaluation metrics and some issues with the method, the qualitative results presented are strong and well-supported. I would vote for acceptance to the workshop.

---

### Decision · Program_Chairs · 2025-03-25

**Decision:**

Accept

**Comment:**

The paper proposes a system for 3D editing using single-shot multi-view diffusion combined with Gaussian Splatting, introducing MV Instruct-Pix2Pix-XL to ensure consistent multi-view image generation. Reviewers commend the qualitative results for their practical effectiveness, coherence, and alignment with textual input. Although concerns were raised regarding the limited novelty due to reliance on existing methods, insufficient evaluation metrics, and potential issues with complexity and inference speed, the overall practical utility and quality of the results are clear strengths.

Given the reviewers' consensus on the method's effectiveness and value to the community, the paper is clearly accepted. Authors are encouraged to clarify evaluation methods, elaborate on model selection and robustness, and provide detailed explanations of the super-resolution step in their camera-ready submission.